# Presence of Non-Diabetic Kidney Diseases in Biopsy Specimens of Diabetic Patients’ Single Center Experience

**DOI:** 10.3390/ijms241914759

**Published:** 2023-09-29

**Authors:** Aleksandar Janković, Nada Dimković, Verica Todorov-Sakić, Ana Bulatović, Nikola Simović, Petar Đurić, Radomir Naumović

**Affiliations:** 1Clinical Department for Nephrology, University Medical Center Zvezdara, 11000 Belgrade, Serbia; todorovverica@yahoo.com (V.T.-S.); ana.milenic@gmail.com (A.B.); simovicnikola84@gmail.com (N.S.); djuricmed@gmail.com (P.Đ.); radomirnaumovic450@gmail.com (R.N.); 2Academy of Medical Sciences of the Serbian Medical Society, 11000 Belgrade, Serbia; dim@eunet.rs; 3School of Medicine, Belgrade University, 11000 Belgrade, Serbia

**Keywords:** type 2 diabetes mellitus, renal biopsy, glomerulonephritis, proteinuria

## Abstract

The complications of type 2 diabetes mellitus (T2DM) are well known and one of them is diabetic chronic kidney disease (DCKD). Over time, it has become clear that patients with T2DM can have nondiabetic chronic kidney diseases (NDCKD), especially those that affect the glomeruli. Clinical indicators for identifying DCKD from NDCKD with high sensitivity and specificity have not yet been identified. Therefore, kidney biopsy remains the golden standard for DCKD diagnosis in patients with T2DM. Despite some indications for kidney biopsy, criteria for a biopsy differ between countries, regions, and doctors. The aim of the study was to analyze the biopsy findings in our T2DM population and the justification of the biopsy according to widely accepted criteria. This single center retrospective study analyzed data from 74 patients with T2DM who underwent kidney biopsy from January 2014 to January 2021. According to the biopsy data, we categorized31 patients in the DN group, patients with typical diabetic glomerulopathy, 11 patients in the mixed group, patients who had pathohistological elements for both DN and non-DN glomerulopathy, and 32 patients in the non-DN group, patients with primary glomerulopathy not linked with DM. In the non-DN and mixed groups, the most frequent glomerulopathy was mesangioproliferative glomerulonephritis, including IgA and non-IgA forms, found in 10 patients, and membranous nephropathy (MN) in 10 patients. We analyzed several parameters and only the amount of proteinuria was found to be significantly linked to biopsy findings related to DN. With the existing criteria for kidney biopsy, we managed to detect changes in the kidneys in about half of our patients with T2DM. These patients required specific treatment, different from that which we use for DCKD patients.

## 1. Introduction

The prevalence of diabetes mellitus (DM) has increased over the past few decades, with projection that this number will be even greater. Namely, according to the World Health Organization, at the moment, about 422 million persons worldwide have DM [1]. In terms of pathology, structural changes in patients with type 2 diabetes mellitus (T2DM) are more heterogeneous than those in type 1 diabetes mellitus (T1DM) patients and they correlate less with clinical manifestations, which are also highly heterogeneous [2,3]. Given this context, studying T2DM patients may have a greater clinical significance.

The complications of T2DM are well known and one of them is diabetic chronic kidney disease (DCKD). An initial finding in DCKD is albuminuria over 300 mg/day with progressive deterioration and consequently impaired kidney function [4]. DCKD is often accompanied by diabetic retinopathy and hypertension, although the absence of diabetic retinopathy does not exclude DCKD [4]. A previously used term for this DM complication was diabetic nephropathy (DN), but now it is used for specific underlying renal pathology [5]. In DCKD linked with T2DM, there are a variety of clinical features. Namely, 3% de novo T2DM patients have albuminuria at the time of diagnosis and a third of patients with diagnosed DCKD do not have diabetic retinopathy [6,7].

Over time, it has become clear that patients with T2DM can have other kidney diseases (nondiabetic chronic kidney diseases—NDCKD), especially those that affect the glomeruli. NDCKD manifests a wide spectrum of pathological lesions and variable prevalence across the world. Regrettably, clinical indicators for identifying DCKD from NDCKD with high sensitivity and specificity have not yet been identified. Therefore, kidney biopsy remains the golden standard for DCKD diagnosis in patients with T2DM. Given that proteinuria and a progressive decline in kidney function are found in diabetic as well as other glomerular diseases, the question arises as to the indications for kidney biopsy in patients with diabetes.

Some indications for kidney biopsy in DM patients are rapid deterioration of kidney function over 5 mL/min per year; severe proteinuria and/or nephrotic syndrome; active urinary sediment (microhaematuria, dysmorphic erythrocyte, and erythrocyte casts); the absence of diabetic retinopathy; the presence of some clinical features specific for other conditions (for example, connective tissue diseases or HIV); positive familiar anamnesis for non-diabetic glomerulopathy [8,9,10]. However, the criteria for a biopsy remain in the domain of the doctor’s decision, and there is not always agreement about this issue. Needless to mention that proper histological diagnosis is of a great importance due to different treatment approaches and overall prognosis. In addition, early identification and treatment of NDCKD is of great significance to reduce global ESRD prevalence and its various complications, such as cardiovascular diseases. Therefore, in this study we analyzed clinical and pathohistological data of patients who had T2DM and underwent kidney biopsy to determine the justification of the biopsy according to the mentioned criteria.

## 2. Results

During the observed period, 74 patients with T2DM underwent kidney biopsy. The patients were predominantly male (68%), the mean age at the time of biopsy was 58 ± 11 years, and the mean duration of DM before biopsy was 6.3 ± 5.9 years. Most had haematuria (74%) and the mean 24-h proteinuria was 8.4 ± 5.5 gr/24 h. Out of all the patients, for 44 we obtained data about the presence of retinopathy, of which 30 (68%) had no signs of this complication. General data about these patients are presented in Table 1.

Out of all the patients, 31 (42%) had typical pathohistological findings for DN, in 32 (43%) patients the findings were not related to DN, and 11 (15%) patients had elements of both DN and some other primary glomerulopathy. In the group of patients with non-DN, the most frequent glomerulopathy was mesangioproliferative glomerulonephritis, including IgA and non-IgA forms, found in nine patients, followed by membranous nephropathy (MN) in six patients (25%). In the mixed group, four (36%) patients had elements of both DN and MN. Pathohistological findings are presented in Table 2.

In Table 3, we present difference in variables among three groups. Analysis has revealed that in among all examined variables there were statistically significant differences in the type of therapy for T2DM. Namely, in the non-DN group, most of the patients were on oral antidiabetic drugs, while in the DN group most of the patients were on insulin therapy. Also, among patients for whom we have known retinopathy status, it is registered that in the non-DN group only three patients had retinopathy and in the mixed group no patients had retinopathy.

According to multivariant binary logistic regression, only proteinuria have shown statistically significant relationship with non-DN biopsy findings (HR 0.89, 95% CI 0.81–0.99, *p* = 0.032). After adjustment for age, gender, and duration of DM, in addition to proteinuria, increases in proteinaemia have also shown a statistically significant relationship with non-DM findings in biposy specimens (Table 4).

## 3. Discussion

The major finding of our study was that about half of our patients with T2DM had either additional glomerulopathy or only non-diabetic kidney disease. In an attempt to more closely determine the parameter that indicates non-diabetic kidney damage, we analyzed several parameters and only the amount of proteinuria was found to be significantly linked to biopsy findings related to DN.

A review that included the results of the 40 studies on the renal biopsy results and pathological NDCKD lesions in T2DM patients showed that the prevalence rate of DN alone ranges from 8.2 to 62.7%, with an average of 41.3% [11]; the prevalence of isolated NDCKD ranges from 0 to 68.6%, with an average of 40.6%; the prevalence of DN plus NDCKD ranges from 0 to 45.5%, with an average of 18.1% [11]. Such differences in results certainly require a detailed analysis of the biopsy criteria as well as the difference between certain geographical regions. Despite the numerous criteria listed with the aim of making it easier to set the indication for kidney biopsy in patients with diabetes [8,9,10], the decision between the centers is quite different. The criteria for renal biopsy are not the same in each region, each country, or even in each nephrologist’s practice [11]. These differences are easy to explain by marked geographical differences in the incidence of T2DM and probably its complications. Most of the countries differ in treatment policies, modern drug availability, patients’ food habits and diabetes control, and use of concomitant medication that may protect kidney function (such as ACEi, ARB, or SGLT2 receptor inhibitors). Therefore, we considered it important to have local results and to adapt well-known criteria with local ones.

Looking at the region, our data are similar to that presented by Horvatic et al. who analyzed 80 Croatian patients with T2DM. Out of all the patients, 46.25% had DN, non-diabetic renal disease superimposed on diabetic nephropathy in 17.5% of patients, and isolated non-diabetic renal disease was found in 36.25% of the patients. The most common non-diabetic renal diseases were also very similar: membranous nephropathy, followed by IgA nephropathy and focal segmental glomerulosclerosis [12]. In one analysis from Bosnia and Herzegovina, kidney biopsy was performed in 17 patients with T2DM and in six (35.3%) NDCKD was found, three (17.6%) had NDCKD superimposed with the diabetic nephropathy, and eight (47.1%) had diabetic nephropathy. Of the patients who had NDCKD, three had membranous nephropathy, one had focal segmental glomerulosclerosis, and two had hypertensive nephroangiosclerosis [13]. The most similar results to ours are from Czech Republic; in their 163 patients with T2DM, 42.3% had DN, 47.3% had NDRD, and 10.4% had both. But in their cohort, IgA nephropathy was the most common primary glomerulopathy [14].

The type of NDCKD disease is also important. According to data from the literature, the most common isolated NDKD pathological type is membranous nephropathy in Asia, Africa (specifically Morocco and Tunisia) and Europe, representing 24.1%, 15.1%, and 22.6% of cases, respectively. In contrast, focal segmental glomerular sclerosis is reported to be the primary pathological type in North America (specifically the USA) and Oceania (specifically New Zealand), representing 22% and 63.9% of cases, respectively, probably due to a larger number of Afro-Americans. Tubulointerstitial disease accounts for a high rate in the mixed group (21.7%), with acute interstitial nephritis being the most prevalent (9.3%), followed by acute tubularnecrosis (9.0%) [11]. Our data show that the most frequent glomerulopathy was mesangioproliferative glomerulonephritis, including IgA and non-IgA forms, found in nine patients, followed by membranous nephropaty (MN) in six patients (25%). Interestingly, we did not find tubulointerstitial nephritis in our patients. Knowing the histological form of NDCKD enables timely and appropriate (most often immunosuppressive) therapy with the aim of reducing the epidemic of kidney failure with all cardiovascular complications.

Determining clinical predictors of diabetic and non-diabetic kidney damage could be of diagnostic help. The amount of proteinuria was the most reliable parameter for DCKD in our population. Multiple clinical parameters including duration of diabetes, presence of diabetic retinopathy and level of proteinuria were used to differentiate DN from NDCKD [15,16]. Classically, a long duration of diabetes (>10 years), the presence of retinopathy, and severe proteinuria strongly suggest DKD [17,18]. Indeed, our traditional knowledge regarding the onset and progression of diabetic nephropathy implied the presence and level of albuminuria and then proteinuria. However, observational study suggests that the level of proteinuria does not discriminate between DN and NDCKD, and that proteinuria is a poor predictor of the type of nephropathy in type 2 diabetes [19]. Similarly, recent evidence has shown that a significant number of diabetes patients had non-albuminuric DKD [20]. This further indicates the need to find new markers that would distinguish between DKD and NDCKD, including kidney biopsy. Hypoproteinemia could be an indicator of extensive proteinuria, malnutrition, extrarenal loss, and/or catabolism [21]. As multivariate binary logistic regression analysis showed that 24-h proteinuria is a significant predictor of non-DN kidney disease, it was expected that proteinaemia would follow that trend. This means that patients with a lower degree of proteinuria and higher proteinemia values lead to the suspicion of non-DN kidney disease.

For a long time, retinopathy was considered the main indicator of microangiopathic changes in T2DM, including nephropathy. Unfortunately, we did not have data on retinal analysis for all our patients but even so, in our group of patients, diabetic retinopathy was more frequent in patients who had DN on histology. Studies have shown that the connection between DN and retinopathy is not always obvious. Still, retinopathy was not confirmed as a predictor of DN by multivariate analysis. The others confirmed that the presence of diabetic retinopathy strongly suggests DKD, and the absence of retinopathy is a major indicator to predict NDKD [22,23,24]. In the study by Castellano et al., the presence of retinopathy had a predictive value of 100% for DN. However, in meta-analysis by Liang et al., 23.6% of patients with biopsy-proven DKD did not have diabetic retinopathy [25,26]. Recent evidence does not agree with the concept that the mere absence of diabetic retinopathy excludes the possibility of NDCKD. Namely, studies have shown a high proportion (50–70%) of DN in patients who did not have diabetic retinopathy [7,15,16].

It is known that hematuria is an atypical finding in patients with T2DM unless there is an additional non-diabetic kidney disease. Despite this, its frequency did not differ between the studied groups (DKD and NDCKD) and hematuria did not prove to be a significant predictor of NDCKD in the studied population. The reason for this is not clear, but it is possible that hematuria can be a finding in (unregulated) hypertension. Also, a recent report of biopsy-proven diabetic nephropathy showed that patients with advanced diabetic nephropathy were accompanied by a high prevalence of hematuria [27]. Another study confirmed that patients with biopsy-proven diabetic nephropathy and hematuria had more advanced pathological findings than those without hematuria, and the presence of hematuria is a significant risk factor for ESKD in patients with diabetic nephropathy [28]. Despite the above mentioned controversies, hematuria was an important criterion for kidney biopsy in our study population.

This paper has some limitations. Single center and retrospective design may not provide data like a well-designed prospective study. The number of patients could be much greater with multicenter research. However, the knowledge gained from this study is very useful to us; with the existing criteria for kidney biopsy, we managed to detect changes in the kidneys in about half of our patients with T2DM. These patients require specific treatment, different from that which we use for DCKD patients. Our future task is to take a closer look at the biopsy criteria with the aim of reducing the number of overlooked patients to a minimum.

## 4. Methods and Materials

We retrospectively analyzed data from patients with T2DM who underwent kidney biopsy from January 2014 to January 2021 at our department. Data about gender, age, kidney biopsy indications, DM duration at the moment of biopsy, urine analysis, biochemical parameters, and 24-h proteinuria were obtained from medical records.

After renal biopsy, all specimens were analyzed by light and immunofluorescence microscopy within 24 h. Typical criteria for DCKD diagnosis included diffuse or nodular glomerulosclerosis, tubulointerstitial fibrosis and atrophy, and variable degrees of hyaline arteriolosclerosis and arterial sclerosis [29]. Proteinuria was measured in 24 h urine using the biuret method [30].

We were adherent to the previously mentioned criteria for renal biopsy [8,9,10].

According to the renal biopsy data, we created three groups of patients: DN group, patients with typical diabetic glomerulopathy; mixed group patients who had pathohistological elements for both DN and non-DN glomerulopathy; non-DN group, patients with primary glomerulopathy not linked with DM.

Statistical analysis was performed using SPSS 22.0. The Kolmogorov–Smirnov test was used to test normal distribution. A one-way ANOVA test, Kruskal–Wallis test, χ^2^ test or Fisher exact test were used to compare variables depending on normality and type of data. Binary logistic regression was performed to analyze the relationship between the presence of non-DN in biopsy specimens and all baseline variables.

## Figures and Tables

**Table 1 ijms-24-14759-t001:** Demographic and clinical data of patients with DM who underwent kidney biopsy.

	Patients with DM (No = 74)
Gender (M/F)	50/24 (68%/32%)
Age (years)	58 ± 11 (min 22–max 84)
DM duration (years)	6.3 ± 5.9 (min 1–max 27)
HbA1c (%)	6.45 ± 1.28 (min 5.1–max 11.1)
DM therapy at the moment of biopsy:	
Insulin therapy	26 (35%)
Oral antidiabetics	32 (43%)
Hygienic-dietery regimen	13 (18%)
Insulin and oral antidiabetics	3 (4%)
Indication for renal biopsy:	
Nephrotic syndrome	23 (31%)
Proteinuria	16 (22%)
Worsening of kidney function and proteinuria	22 (30%)
Suspicion on SAD	13 (17%)
Retinopathy (yes/no/NA)	30/14/30 (40.5%/19%/40.5%)
Haematuria (yes/no)	55/19 (74%/26%)
Albuminaemia (g/L)	30 ± 9 (min 13–max 48)
Proteinaemia (g/L)	64 ± 10 (min 40–max 81)
Quantitative proteinuria (gr/24 h)	8.4 ± 5.5(min 0.10–max 25.00)
Cholesterol (mmol/L)	5.8 ± 2.1 (min 2.6–max 13.0)
Thrygliceride (mmol/L)	2.6 ± 1.4 (min–max)
Creatinine (mcmol/L)	222 ± 188 (min 54–max 1246)

SAD: systematic autoimmune disorder.

**Table 2 ijms-24-14759-t002:** Pathohistological findings in non-DN group and mixed group of patients.

	Non-DN Group No = 32	Mixed Group (DN + Non-DN), No = 11
MesPGN	9 (28.1%)	1 (9%)
MN	6 (18.6%)	4 (36%)
FSGS	3 (9.4%)	3 (28%)
RPGN	2 (6.2%)	/
MCD	1 (3.2%)	/
MPGN	1 (3.2%)	1 (9%)
amyloidosis	1 (3.2%)	/
other	9 (28.1%)	2 (18%)

FSGS—foscal-segmental glomerular sclerosis; MCD—minimal change disease; MesPGN—mesangioproliferative glomerulonephritis including IgA and non IgA forms; MN—membranous nephropathy; MPGN—membranoprilferative glomerulonephritis; RPGN—rapidly progressive glomerulonephritis.

**Table 3 ijms-24-14759-t003:** Demographic and clinical data of patients from three different pathohistological groups.

	DN Group	Non-DN Group	Mixed Group (DN + Non-DN)	*p* *
No(%) of patients	31 (42)	32 (43)	11 (15)	
Gender (M/F)	20/11	22/10	8/3	0.867
Age (years)	57 ± 14	60 ± 7	57 ± 12	0.401
HbA1c (%)	6.83 ± 1.60	6.29 ± 0.95	5.88 ± 0.84	0.078
DM therapy at the moment of biopsy:				
Insulin therapy	15 (48%)	9 (28%)	2 (18%)	0.011
Oral antidiabetics	7 (23%)	16 (50%)	9 (72%)	
Hygienic-dietery regimen	6 (19%)	7 (22%)	0	
Insulin and oral antidiabetics	3 (10%)	0	0	
Indication for renal biopsy:				
Nephrotic syndrome	8 (26%)	10 (31%)	5 (46%)	0.815
proteinuria	6 (19%)	8 (25%)	2 (18%)	
Worsening of kidney function and proteinuria	12 (39%)	8 (25%)	2 (18%)	
Suspicion on SAD	5 (16%)	6 (19%)	2 (18%)	
DM duration (years)	5.7 ± 6.0	7.7 ± 6.2	4.1 ± 2.8	0.142
Retinopathy (yes/no/NA)	(11/9/11)	(3/14/15)	(0/7/4)	0.024
Haematuria (yes/no)	23/8	23/9	9/2	0.809
Albuminaemia (g/L)	29 ± 7	31 ± 9	32 ± 10	0.754
Proteinaemia (g/L)	61 ± 8	65 ± 11	63 ± 13	0.388
24 h proteinuria (gr/24h)	9.15 ± 4.93	7.01 ± 5.35	10.15 ± 6.72	0.112
Cholesterol (mmol/L)	5.7 ± 2.1	5.6 ± 1.9	6.6 ± 2.6	0.399
Triglyceride (mmol/L)	2.3 ± 1.2	2.6 ± 1.0	3.5 ± 2.4	0.244
Creatinine (mcmol/L)	238 ± 160	223 ± 226	178 ± 139	0.412

* According to one-way ANOVA test, Kruskal–Wallis test, χ^2^ test or Fisher exact test used where apropriate.

**Table 4 ijms-24-14759-t004:** Multivariant binary logistic regression as a prediction for non-DN biopsy finding.

	*p*	OR	95% CI
24-h proteinuria	0.013	0.705	0.534–0.930
Proteinaemia	0.035	1.114	1.008–1.231

## Data Availability

Not applicable.

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
