# Peer review of "Presence of Non-Diabetic Kidney Diseases in Biopsy Specimens of Diabetic Patients’ Single Center Experience"

_ijms, 2023, doi:10.3390/ijms241914759_

Round 1

Reviewer 1 Report

In this study, the authors aimed to analyze biopsy findings in patients with T2DM. This is a single-center, retrospective study including 74 patients with T2DM who underwent kidney biopsy.

The study is clinically of interest. The study provides useful information for readers. However, there are some critiques.

1.        Background of patients should be described more precisely. Information regarding glycemic control, i.e., HbA1c, and anti-diabetic medications should be listed in the table.

2.        As the authors mention, the lack of information on diabetic retinopathy is a major limitation. Complications other than nephropathy, i.e., retinopathy, neuropathy and CVD should be assessed before renal biopsy, as these information are very useful to predict the presence of diabetic nephropathy. This point should be more clearly discussed in the manuscript.

3.        Presence of hematuria is another important predictor for non-diabetic kidney disease. This point should be more precisely assessed and discussed.

4.        The authors stated that amount of proteinuria was found to be linked to biopsy finding related to diabetic nephropathy. This should be an important message of this study. Please explain the clinical meaning of this finding more clearly for the readers.

Some minor typos should be edited.

Round 2

Reviewer 1 Report

The authors responded to the comments properly.

Reviewer 2 Report

The work was surely improved, and the results about proteinuria have been  discussed,  but my question about proteinemia was not answered: still proteinemia is in the results but not in discussion.

Almost some comment is needed. 
